# Public perceptions of Ebola vaccines and confidence in health services to treat Ebola, malaria, and tuberculosis: Findings from a cross-sectional household survey in Uganda, 2020

**Aybüke Koyuncu**[1]*, **Rosalind J. Carter**[2], **Joseph Musaazi**[3], **Apophia Namageyo-Funa**[1], **Victoria M. Carter**[1], **Mohammed Lamorde**[3], **Dimitri Prybylski**[1], **Rose Apondi**[1], **Tabley Bakyaita**[4], **Amy L. Boore**[1], **Jaco Homsy**[1,5], **Vance R. Brown**[1], **Joanita Kigozi**[3], **Maria Sarah Nabaggala**[3], **Vivian Nakate**[3], **Emmanuel Nkurunziza**[3], **Daniel F. Stowell**[1], **Richard Walwema**[3], **Apollo Olowo**[3], **Mohamed F. Jalloh**[1]

**1** Center for Global Health, U.S. Centers for Disease Control and Prevention, Atlanta, Georgia, United States of America, **2** National Center for Immunization and Respiratory Diseases, U.S. Centers for Disease Control and Prevention, Atlanta, Georgia, United States of America, **3** Infectious Diseases Institute, College of Health Sciences, Makerere University, Kampala, Uganda, **4** Uganda Ministry of Health, Kampala, Uganda, **5** Institute for Global Health Sciences, University of California San Francisco, San Francisco, California, United States of America

* akoyuncu@berkeley.edu

## Abstract

Uganda used Ebola vaccines as part of its preparedness and response during the 2018–2020 10th Ebola virus disease (EVD) outbreak in neighboring Democratic Republic of the Congo (DRC). We evaluated the public's perceptions of Ebola vaccines and compared their confidence in health services to treat Ebola versus malaria and tuberculosis as part of a survey on Ebola knowledge, attitudes, and practices (KAP) conducted in March 2020. A cross-sectional household survey was implemented in six districts in Uganda using multi-stage cluster sampling to randomly select participants. The districts were purposively selected from districts classified by the government as at high- or low-risk for an EVD outbreak. We describe perceptions of Ebola vaccines and confidence in health services to treat Ebola, tuberculosis, and malaria. Modified Poisson regression modeling was used to identify the demographic correlates of these outcomes. Among 3,485 respondents, 18% were aware of Ebola vaccines. Of those, 92% agreed that the vaccines were needed to prevent Ebola. Participants aged 15–24 years were 4% more likely to perceive such need compared to those 60 years and older (adjusted prevalence ratio [aPR] 1.04, 95% confidence interval [CI] 1.0–1.08). The perceived need was 5% lower among participants with at least some secondary education compared to uneducated participants (aPR 0.95; 0.92–0.99). Overall, 81% of those aware of the vaccines believed that everyone or most people in their community would get vaccinated if offered, and 94% said they would likely get vaccinated if offered. Confidence in health services to treat Ebola was lower compared to treating malaria or tuberculosis (55% versus 93% and 77%, respectively). However, participants from the EVD

**Data Availability Statement:** The de-identified dataset and codebook can be found here: https://github.com/akoyuncu4/evd_uga_2020.git.

**Funding:** The Center for Global Health at the U.S. Centers for Disease Control and Prevention funded the Infectious Disease Institute in Kampala to implement the survey (Cooperative Agreement # NU2GGH001744). The funders had no role in study design, data collection and analysis, decision to publish, or preparation of the manuscript.

**Competing interests:** The authors have declared that no competing interests exist.

high-risk districts were 22% more likely to be confident in health services to treat Ebola compared to those in low-risk districts (aPR: 1.22; 95% CI: 1.08, 1.38). Our findings suggest that intent to take an Ebola vaccine during an outbreak was strong, but more work needs to be done to increase public awareness of these vaccines. The public's high confidence in health services to treat other health threats, such as malaria and tuberculosis, offer building blocks for strengthening their confidence in health services to treat EVD in the event of an outbreak.

## Introduction

Outbreaks of Ebola virus disease (EVD) are complex and require a comprehensive package of interventions. These include case management, infection prevention, and control, surveillance and contact tracing, safe and dignified burials, and risk communication and community engagement (RCCE) [1]. Additionally, the World Health Organization (WHO) 2018 Strategic Advisory Group of Experts (SAGE) on Immunization recommended rVSV-ZEBOV-GP (Merck) vaccine against EVD as a safe and effective prevention and response tool in EVD outbreaks caused by Zaire ebolavirus [2].

The 10th EVD outbreak in the Democratic Republic of Congo (DRC) was the second largest ever after the 2014–2016 West Africa epidemic and was the deadliest in DRC history [3]. It lasted nearly 2 years and resulted in over 3600 cases, with a case fatality rate of 66% [3]. In 2018, Uganda became the first country to use Ebola vaccines as part of preparedness efforts in response to the outbreak in DRC [4]. The Uganda Ministry of Health (MOH), with support from the World Health Organization (WHO) and other partners, offered the single dose Ebola vaccine, rVSV-ZEBOV, manufactured by Merck, on a compassionate use basis to healthcare and frontline response workers in Ugandan health facilities bordering DRC. At the time of implementation, the unlicensed Merck product was the only vaccine approved for use in an outbreak and for disease prevention among frontline health workers at high risk of exposure. From November 2018 through April 2019, 4419 healthcare workers were vaccinated in 13 high-risk districts [4]. In June 2019, three imported cases of Ebola in Kasese District, Uganda, were confirmed in a family traveling from DRC [5]. In response, the MOH and WHO offered the Merck Ebola vaccine using a ring vaccination approach to contacts, including both community members and healthcare workers [5]. In August 2019, another imported case of EVD was confirmed in Kasese District, and vaccination activities were implemented. However, neither instance of EVD importation resulted in additional spread of the virus in Uganda.

To increase vaccine acceptance and help contain and end outbreaks, community engagement strategies used during Ebola preparedness and response efforts must build the public's confidence in Ebola vaccines by creating awareness of Ebola vaccines and promoting their need, benefits, and safety [6]. Public perceptions about Ebola vaccines have been studied in various settings, and results have consistently shown high acceptability, perceived need, and future intention to accept, and acceptance of Ebola vaccines [7–12]. For example, 99% (95% CI: 98.5, 99.4) of healthcare workers offered Ebola vaccines in Eastern DRC self-reported accepting the vaccine during the 2018–2020 EVD outbreak, although first-offer uptake was significantly lower (70%; 95% CI: 67.1, 73.5) and many healthcare workers did not accept the vaccine until it was offered to them multiple times [11]. Notably, most existing evidence on the public perception of Ebola vaccines has been collected in countries involved in the West African Ebola virus epidemic or 2018–2020 outbreak in eastern DRC, and some studies have taken

place during ongoing EVD outbreaks [9, 12]. Perceptions about Ebola vaccines may differ in settings such as Uganda where outbreaks have historically been less widespread and a smaller proportion of the population may have personal experiences related to EVD. In Guinea, for example, intention to accept an Ebola vaccine towards the end of the 2014–2016 West African Ebola virus epidemic was significantly higher among individuals who were aware of new Ebola transmissions in their community and/or knew someone affected by Ebola [9].

Vaccine confidence is defined as trust in (i) the effectiveness and safety of vaccines, (ii) the system that delivers them, including the reliability and competence of the health services and health professionals, and (iii) the motivations of policy-makers who decide on the need of vaccines [13]. In addition to vaccine confidence, vaccine perceptions and uptake are associated with risk perception of disease, structural and psychological constraints (e.g. vaccine access and convenience), individual cost-benefit calculations, social norms, and collective responsibility to protect others [14, 15]. Confidence in health systems and the services they offer also influence perceptions about vaccines, and vaccine confidence is correlated with confidence in the health system [16]. Interventions that leverage trust in healthcare systems (e.g. vaccination recommendations from trusted providers) are among the most effective ways to increase vaccine uptake [15]. Understanding vaccine confidence and trust in health services is thus a key part of building successful community engagement strategies aimed at strengthening Ebola preparedness.

Ebola vaccines are a pillar of EVD prevention and response, but the limited supply of vaccines, and therefore access to vaccines precludes their widespread use. Community engagement strategies must therefore build trust in vaccines despite the ineligibility of most individuals for vaccination during an outbreak. To guide future RCCE strategies in EVD outbreak preparedness and response in Uganda, we evaluated the public's perceptions of Ebola vaccines and compared their confidence in health services to treat Ebola versus malaria and tuberculosis.

## Materials and methods

We conducted a cross-sectional household survey among persons 15 years of age and older in six districts in Uganda following the 2018–2020 10th EVD outbreak in neighboring DRC. Our purposive selection of these six districts was informed by the Uganda government's classification of districts into varying risk profiles for a potential cross border spillover EVD outbreak from the DRC [4]. We selected four high-risk districts (Arua, Kasese, Kisoro, and Greater Kampala) and two low-risk districts (Busia and Lamwo) for the survey, the methods of which have been elaborated elsewhere [17]. To summarize, we used multistage sampling to select geographic clusters, households, and individuals to interview. The 2016 Uganda National Population Housing Census served as the sampling frame for the random selection of clusters with probability proportional to size within each district [18]. Sample size was calculated to provide district-level estimates of the public's knowledge, beliefs, attitudes and practices related to EVD risks. After adjusting for an expected 25% level of non-responsiveness, 640 eligible individuals from 20 clusters were approached for consent in each district. Trained interviewers administered questionnaires to participants in local languages.

### Measures

Our main outcomes were perceptions of Ebola vaccines and confidence in health services to treat Ebola, malaria, and tuberculosis. Perceptions of Ebola vaccines were only assessed among respondents who reported being aware of Ebola vaccines. Awareness was assessed by asking participants if they had ever heard about an Ebola vaccine before their interview.

**Perceptions of Ebola vaccines.** We used three variables to measure perceptions of Ebola vaccines: (1) perceived need for Ebola vaccines to prevent EVD during an EVD outbreak, (2) perceived acceptability of Ebola vaccines by others if offered in the event of an outbreak, and (3) intention to accept the vaccine for oneself if offered in the event of an outbreak. These variables were adapted because they showed acceptable scale reliability and criterion validity in Sierra Leone during the 2014–2016 Ebola epidemic [12]. Questions on perceptions of Ebola vaccines were only asked to individuals who reported having heard of the vaccines prior to their interview. Perceived need for Ebola vaccines was measured by asking participants if they agree, somewhat agree, or disagree with the statement that "If Uganda started having cases of Ebola, an Ebola vaccine is needed to help prevent the spread of the disease in the country." We generated a binary variable to group participants who said they agree or somewhat agree (coded 1) versus disagree (coded 0). To measure perceptions of acceptability by others in the community, participants were asked "If there is an Ebola outbreak in your district, how many people in your community do you think would agree to take an Ebola vaccine if they were offered it?". A binary variable was created to group participants who said everyone or most people would agree (coded 1) or versus some or no one would agree (coded 0). Intention to accept Ebola vaccines was measured by asking participants "If there is an Ebola outbreak in your district, how likely would you be to take an Ebola vaccine for yourself if you were offered it?". We generated a binary variable where individuals who were very likely or somewhat likely to take an Ebola vaccine were grouped separately (coded 1) from individuals who were not very likely or not at all likely (coded 0).

**Confidence in health services.** Confidence in the health care system was measured by asking participants to rate their confidence in the system to care for Ebola, tuberculosis (TB), and malaria. Questions on confidence in the health care system were asked to all survey participants. Participants had the option to rate their confidence as "not at all confident," "somewhat confident," and "very confident" in the district health services to treat each of the three diseases. We generated separate binary categorical variables for each disease, and individuals who were very or somewhat confident were categorized as having high confidence (coded 1) while individuals who were not at all confident were categorized as having low confidence (coded 0).

## Statistical analysis

All analyses were conducted in STATA 16 (College Station, Texas) and weighted to account for the complex survey design. District-specific post-stratification weights were used to pool data from all districts included in this analysis. We conducted bivariable analyses by comparing frequencies, proportions, and 95% confidence intervals (CI) of proportions of our outcomes of interest across sociodemographic characteristics, and residence in a high-risk versus low-risk district.

We then used multivariable models to explore associations between sociodemographic characteristics and our main outcomes. The following sociodemographic covariates were included in all models: sex (females *vs* male), head of household status (no *vs* yes), age category (60+ *vs* 45–59, 35–44, 25–34, 15–24 years), religion (Christian *vs* Muslim), education (no education *vs* some primary *or* at least some secondary) residential setting (urban *vs* rural), district (low Ebola risk *vs* high Ebola risk).

Models for Ebola vaccine perceptions were restricted to individuals who were aware of Ebola and Ebola vaccines. Separate multivariable models were used to examine the associations of sociodemographic covariates with: (1) the perceived need for Ebola vaccines, (2) perceived community acceptability of Ebola vaccines, and (3) intention to accept Ebola vaccines. Separate multivariable models were used to examine the associations of sociodemographic

covariates with: (1) confidence in health services to treat EVD, (2) confidence in health services to treat malaria, and (3) confidence in health services to treat tuberculosis. Models excluded individuals who had never heard of a respective health outcome (EVD, malaria, tuberculosis) prior to their interview.

We fit the models using modified Poisson regression with robust variance estimation to calculate the adjusted prevalence ratios and corresponding 95% confidence intervals (CI). We used variance inflation factor (VIF) to examine multicollinearity in each of the multivariable models [19]. No evidence of multicollinearity was detected in any of the models (mean VIF values were < 4 for each of the six models). In all models, a two-sided $P$ value < 0.05 was considered statistically significant. Data missingness was < 2%, therefore we did not perform sensitivity analysis for missing data.

## Ethical considerations

Ethical approval was obtained from the Uganda Virus Research Institute Research Ethics Committee (UVRI REC GC/127/19/11/756) and the survey was registered with the Uganda National Council for Science and Technology (HS531ES). The U.S. Centers for Disease Control and Prevention approved the survey as a non-research public health activity. All respondents provided individual written (or thumb-printed if illiterate) informed consent prior to the interview. For respondents aged 15–17 years, consent was obtained from a parent, guardian, or household head, as well as assent from the respondent. Informed consent and assent forms were translated into local languages. Additional information regarding the ethical, cultural, and scientific considerations specific to inclusivity in global research is included in the Supporting Information (S1 Checklist).

## Results

The study population included 3,485 participants (Table 1). Across all six districts, 60% of participants were female (N = 2,106), 78% were aged 25 years or older (N = 2,724), and 69% lived in rural communities (N = 2,396).

Overall, almost all individuals (96%) had heard of Ebola prior to the interview. Among the individuals that had heard of Ebola (N = 3352), only 627 (18%) had heard of Ebola vaccines prior to their interview. Among 627 participants who had heard of Ebola vaccines, 25 said they had been offered the Ebola vaccine, of whom 18 (72%) were vaccinated, 5 (20%) declined vaccination, and 2 (8%) declined to answer or had missing values.

Among 627 participants who had heard of Ebola vaccines, 88% (95% CI: 83, 92) agreed and 4% (95% CI: 2.5, 6.1) somewhat agreed that a vaccine is needed to help prevent the spread of EVD in the country during an outbreak (Table 2).

Participants aged 15–24 years were 4% more likely to perceive a need for vaccines during an outbreak compared to those 60 years and older (aPR 1.04; 95% CI 1.00–1.08) (Table 3).

The perceived need was 5% lower among participants with at least some secondary education compared to uneducated participants (aPR 0.95; 0.92–0.99). If an Ebola vaccine was offered during an outbreak in their district, 94% (95% CI: 90, 96) were likely to take it and most individuals believed everyone (31%; 95% CI: 27, 36) or most people (50%; 95% CI: 45, 55) in their district would agree to take the vaccine. Perceptions about vaccines did not vary by high versus low-risk districts. Over 75% of participants stated that they had no concerns about the Ebola vaccine. Among those that expressed a concern, fear of vaccine side effects was most frequently reported (9%; 95% CI: 7, 11; Fig 1).

Almost everyone in our survey was aware of EVD, TB and malaria with little variability by sociodemographic characteristics (S1 Table). Awareness about EVD prior to the interview was

**Table 1. Sociodemographic characteristics of participants and district-level Ebola risk profile, Uganda, March 2020.**

| Characteristic | Overall (N = 3,485) | District Ebola risk | |
|---|---|---|---|
| | | Low-risk district (N = 1,227) | High-risk district (N = 2,258) |
| **Sex** | | | |
| Female | 2,106 (60.4) | 693 (56.5) | 1,413 (62.6) |
| Male | 1,379 (39.6) | 534 (43.5) | 845 (37.4) |
| **Head of household** | | | |
| No | 1,716 (49.2) | 611 (49.8) | 1,105 (48.9) |
| Yes | 1,769 (50.8) | 616 (50.2) | 1,153 (51.1) |
| **Age (years)**¶ | | | |
| 15–24 | 761 (21.8) | 284 (23.2) | 477 (21.1) |
| 25–34 | 878 (25.2) | 286 (23.3) | 592 (26.2) |
| 35–44 | 715 (20.5) | 244 (19.9) | 471 (20.9) |
| 45–59 | 686 (19.7) | 242 (19.7) | 444 (19.7) |
| 60 or older | 445 (12.8) | 171 (13.9) | 274 (12.1) |
| **Religion**‖ | | | |
| Christian | 3,184 (91.6) | 1,116 (91.3) | 2,068 (91.8) |
| Muslim | 293 (8.4) | 107 (8.8) | 186 (8.3) |
| **Education**‖ | | | |
| No formal education | 640 (18.4) | 201 (16.4) | 439 (19.5) |
| Some primary | 1,747 (50.2) | 650 (53.1) | 1,097 (48.6) |
| Some secondary or higher | 1,093 (31.4) | 373 (30.5) | 720 (31.9) |
| **Residential setting**¶ | | | |
| Urban | 1,089 (31.3) | 270 (22.0) | 819 (36.3) |
| Rural | 2,396 (68.8) | 957 (78.0) | 1,439 (63.7) |

‖ Missing values: religion (n = 8), education (n = 5)

¶ Age categorized and residential setting classified as per the 2016 Uganda Demographic and Health Survey provided by the Uganda Bureau of Statistics

higher in high-risk versus low-risk districts (99% vs 89%) but did not vary across other socio-demographic characteristics (S1 Table). Strong confidence in health services to treat different diseases was highest for malaria (65%), followed by TB (44%) and EVD (29%) (Table 2).

We identified no major differences in confidence in health services to treat EVD by socio-demographic variables in multivariable analyses (Table 4).

However, individuals living in high-risk districts reported more confidence in health services to treat EVD compared to individuals in low-risk districts (aPR 1.22; 95% CI: 1.08, 1.38) (Table 4). High and low-risk districts had similar confidence in health services to treat TB (aPR: 0.99; 95% CI: 0.93, 1.05) and malaria (aPR: 0.99; 95% CI: 0.97, 1.03).

## Discussion

Among survey respondents who had heard of Ebola vaccines, we found high perceived need for Ebola vaccines and high intent to take an Ebola vaccine in Uganda during a protracted EVD outbreak in neighboring DRC. This finding suggests that messaging on the availability, safety, and effectiveness of Ebola vaccines is a vital part of Ebola preparedness and outbreak response efforts.

Additionally, nearly everyone that was aware of Ebola vaccines intended to accept the vaccine if offered during an outbreak and believed that the vaccine would help prevent the spread of EVD in event of an outbreak. Approximately 8 out of every 10 participants that were aware of the vaccine perceived that all or most people in their community would accept the vaccine if

**Table 2. Perceptions about Ebola vaccines and health services by district-level Ebola risk profile, Uganda, March 2020.**

| | All districts | | Low-risk districts | | High-risk districts | |
|---|---|---|---|---|---|---|
| | N | % (95% CI) | N | % (95% CI) | N | % (95% CI) |
| **EBOLA VACCINE CONFIDENCE** | | | | | | |
| **Importance If Uganda started having cases of Ebola, an Ebola vaccine is needed to help prevent the spread of the disease in the country¶** | | | | | | |
| Agree | 609 | 88.2 (83.4, 91.9) | 109 | 87.5 (76.5, 93.8) | 500 | 88.4 (82.7, 92.4) |
| Somewhat agree | | 4.0 (2.5, 6.1) | | 11.2 (5.1, 22.7) | | 2.5 (1.5, 4.2) |
| Disagree | | 7.8 (4.5, 13.2) | | 1.3 (0.2, 8.1) | | 9.1 (5.1, 15.6) |
| **Acceptability: If there is an Ebola outbreak in your district, how many people in your community do you think would agree to take an Ebola vaccine if they were offered it?¶¶** | | | | | | |
| No one | 611 | 0.5 (0.1, 3.1) | 113 | 0.0 | 498 | 0.7 (0.1, 3.6) |
| Some people | | 18.4 (15.6, 21.5) | | 18.0 (10.2, 29.8) | | 18.5 (15.7, 21.5) |
| Most people | | 49.8 (44.9, 54.7) | | 56.2 (43.9, 67.7) | | 48.6 (43.3, 53.9) |
| Everyone | | 31.2 (26.9, 35.9) | | 25.8 (16.0, 38.8) | | 32.3 (27.6, 37.3) |
| **Intention: If there is an Ebola outbreak in your district, how likely would you be to take an Ebola vaccine for yourself if you were offered it?¶¶¶** | | | | | | |
| Very likely to take it | 617 | 94.0 (90.4, 96.3) | 114 | 93.5 (86.2, 97.0) | 503 | 94.1 (89.8, 96.6) |
| Somewhat likely to take it | | 4.0 (2.5, 6.3) | | 4.0 (1.6, 9.7) | | 4.0 (2.4, 6.7) |
| Not very likely to take it | | 1.0 (0.3, 3.0) | | 1.7 (0.2, 11.8) | | 0.8 (0.2, 3.3) |
| Not at all likely to take it | | 1.0 (0.3, 3.5) | | 0.8 (0.1, 6.2) | | 1.1 (0.3, 4.2) |
| **HEALTH SERVICES TO TREAT EBOLA, MALARIA, AND TUBERCULOSIS** | | | | | | |
| **Total respondents** | 3,485 | | 1,227 | | 2,258 | |
| **EVD: How confident are you in the health services in your district to treat Ebola?‡** | | | | | | |
| Not at all confident | 3,302 | 45.3 (41.8, 48.8) | 1,109 | 51.7 (46.6, 56.8) | 2,193 | 44.0 (40.0, 48.1) |
| Somewhat confident | | 26.1 (23.6, 28.7) | | 32.2 (27.8, 36.9) | | 24.9 (22.1, 27.9) |
| Very confident | | 28.6 (25.0, 32.5) | | 16.1 (12.6, 20.3) | | 31.1 (26.8, 35.7) |
| **TB: How confident are you in the health services in your district to treat tuberculosis (TB/dry cough)?‡‡** | | | | | | |
| Not at all confident | 3,330 | 23.0 (20.5, 25.6) | 1,183 | 16.2 (12.8, 20.4) | 2,147 | 24.3 (21.4, 27.4) |
| Somewhat confident | | 33.4 (30.4, 36.5) | | 52.9 (49.1, 56.7) | | 29.5 (26.1, 33.2) |
| Very confident | | 43.6 (40.0, 47.4) | | 30.9 (27.0, 35.1) | | 46.2 (41.8, 50.6) |
| **Malaria: How confident are you in the health services in your district to treat malaria?‡‡‡** | | | | | | |
| Not at all confident | 3,460 | 7.0 (4.3, 11.3) | 1,216 | 6.4 (4.4, 9.2) | 2,244 | 7.1 (4.0, 12.4) |
| Somewhat confident | | 27.9 (25.3, 30.7) | | 52.0 (46.4, 57.5) | | 23.2 (20.2, 26.4) |
| Very confident | | 65.1 (60.7, 69.2) | | 41.7 (35.9, 47.7) | | 69.7 (64.5, 74.4) |

N denotes number of respondents (un-weighted), % percentage estimates and CI confidence interval both weighted for survey sampling

§ Design-based F-statistic p values for comparing high-risk versus low-risk districts.

†Analysis on confidence on Ebola vaccine included only those who had ever heard about Ebola vaccine prior to the interview, overall = 621 (Low-risk districts n = 115, High-risk districts n = 506)

¶ Analysis excluded: 11 responded don't know

¶¶ Analysis excluded: 10 responded don't know

¶¶¶ Analysis excluded: 3 responded don't know, and 1 missing response

‡ Of the total respondents in the survey (N = 3,485), this analysis excluded: n = 133 responded never heard of Ebola, n = 50 declined to respond.

‡‡ Analysis excluded: those responded never heard of TB (n = 116), declined to respond (n = 39).

‡‡‡ Analysis excluded: those responded never heard of malaria (n = 12), declined to respond (n = 13).

offered during an outbreak. Less than 20% of survey participants had heard of Ebola vaccines prior to their interview, and it is possible that those who were aware of the vaccine were also more likely to have positive attitudes toward the vaccine compared to those who were not. Risk communication materials for the public in Uganda included only basic information about the availability of vaccines to prevent EVD transmission. More detailed information

**Table 3. Perception about Ebola vaccine by sociodemographic characteristics, Uganda, March 2020.**

| Characteristic | Perceived that Ebola vaccines are needed to prevent Ebola [†] | | | Believed all or most people would accept Ebola vaccine if offered[†††] | | | Intended to accept Ebola vaccine if offered[††] | | |
|---|---|---|---|---|---|---|---|---|---|
| | N = 609 | | | n = 611 | | | N = 617 | | |
| | n | % | aPR (95% CI) | n | % | aPR (95% CI) | n | % | aPR (95% CI) |
| **Sex** | | | | | | | | | |
| Female | 332 | 90.1 | Reference | 331 | 79.1 | Reference | 337 | 98.7 | Reference |
| Male | 277 | 96.4 | 0.98 (0.95, 1.00) | 280 | 84.8 | 1.01 (0.92, 1.10) | 280 | 96.6 | 0.99 (0.96, 1.01) |
| **Head of household** | | | | | | | | | |
| No | 256 | 85.3 | Reference | 258 | 83.7 | Reference | 260 | 98.3 | Reference |
| Yes | 353 | 97.2 | 1.01 (0.98, 1.05) | 353 | 79.2 | 1.08 (0.97, 1.20) | 357 | 97.8 | 1.00 (0.97, 1.03) |
| **Age (years)[¶]** | | | | | | | | | |
| 15–24 | 106 | 100 | 1.04* (1.00, 1.08) | 107 | 82.7 | 1.02 (0.89, 1.18) | 107 | 97.0 | 1.01 (0.96, 1.05) |
| 25–34 | 161 | 80.6 | 0.99 (0.96, 1.02) | 163 | 64.7 | 0.91 (0.79, 1.05) | 165 | 97.9 | 0.99 (0.95, 1.03) |
| 35–44 | 134 | 98.6 | 1.00 (0.96, 1.04) | 131 | 92.3 | 1.05 (0.94, 1.18) | 133 | 99.0 | 1.00 (0.97, 1.04) |
| 45–59 | 139 | 96.4 | 0.99 (0.95, 1.03) | 140 | 91.2 | 1.05 (0.95, 1.17) | 142 | 97.3 | 1.01 (0.97, 1.04) |
| 60 or older | 69 | 98.9 | Reference | 70 | 90.7 | Reference | 70 | 98.8 | Reference |
| **Religion** | | | | | | | | | |
| Christian | 553 | 95.3 | Reference | 553 | 79.4 | Reference | 559 | 97.8 | Reference |
| Muslim | 56 | 75.8 | 0.98 (0.92, 1.05) | 58 | 89.7 | 1.06 (0.93, 1.21) | 58 | 99.0 | 0.99 (0.93, 1.05) |
| **Education[‖]** | | | | | | | | | |
| No formal education | 92 | 100 | Reference | 89 | 94.7 | Reference | 91 | 100.0 | Reference |
| Some primary | 275 | 91.3 | 0.98 (0.96, 1.00) | 280 | 82.6 | 0.96 (0.86, 1.07) | 282 | 99.6 | 0.99 (0.98, 1.01) |
| Some secondary or higher | 241 | 91.6 | 0.95* (0.92, 0.99) | 241 | 77.0 | 0.95 (0.84, 1.07) | 243 | 96.1 | 0.97 (0.93, 1.00) |
| **Residential setting[¶]** | | | | | | | | | |
| Urban | 206 | 88.5 | Reference | 205 | 78.7 | Reference | 206 | 97.8 | Reference |
| Rural | 403 | 98.5 | 1.01 (0.98, 1.06) | 406 | 85.1 | 1.06 (0.97, 1.17) | 411 | 98.3 | 1.01 (0.98, 1.05) |
| **District** | | | | | | | | | |
| Low Ebola risk | 109 | 98.7 | Reference | 113 | 82.0 | Reference | 114 | 97.4 | Reference |
| High Ebola risk | 500 | 90.9 | 0.98 (0.95, 1.00) | 498 | 80.9 | 1.07 (0.95, 1.18) | 503 | 98.1 | 0.99 (0.96, 1.03) |

N denotes number of respondents (un-weighted), % row percentage estimates and CI confidence interval both weighted for survey sampling

\* p< 0.05

\*\*p<0.01; Design-based F-statistic p values comparing prevalence across categories of a specific characteristic.

† Analysis performed on N = 609 of 621 respondents who reported that they were aware of Ebola vaccine: n = 1 declined to respond and n = 11 responded don't know.

†† Analysis performed on N = 617 of 621 respondents who reported that they were aware of Ebola vaccine: n = 1 declined to respond and n = 3 responded don't know.

††† Analysis performed on N = 617 of 621 respondents who reported that they were aware of Ebola vaccine: n = 10 responded don't know.

‖ Missing values: education (n = 1)

¶ Age categorized and residential setting classified as per the 2016 Uganda Demographic and Health Survey provided by the Uganda Bureau of Statistics

about the importance, efficacy and safety of vaccines was targeted towards health care and frontline workers who were eligible for vaccination and was not widespread in communities. Awareness of Ebola vaccines was highest (42%) in the outbreak district, Kasese, and lowest in the low-risk districts, Busia (8%) and Lamwo (12%), providing some evidence that vaccine administration was associated with higher community awareness of Ebola vaccines.

Nearly half of all respondents that were aware of Ebola prior to their interview did not have confidence in health services to treat Ebola in their district. In contrast to malaria and tuberculosis treatment, EVD treatment occurs in Ebola Treatment Units (ETUs) that are specifically set up only during EVD outbreaks. The survey was not conducted during an EVD outbreak, therefore the low confidence in routine health services to treat Ebola may reflect lack of ETUs

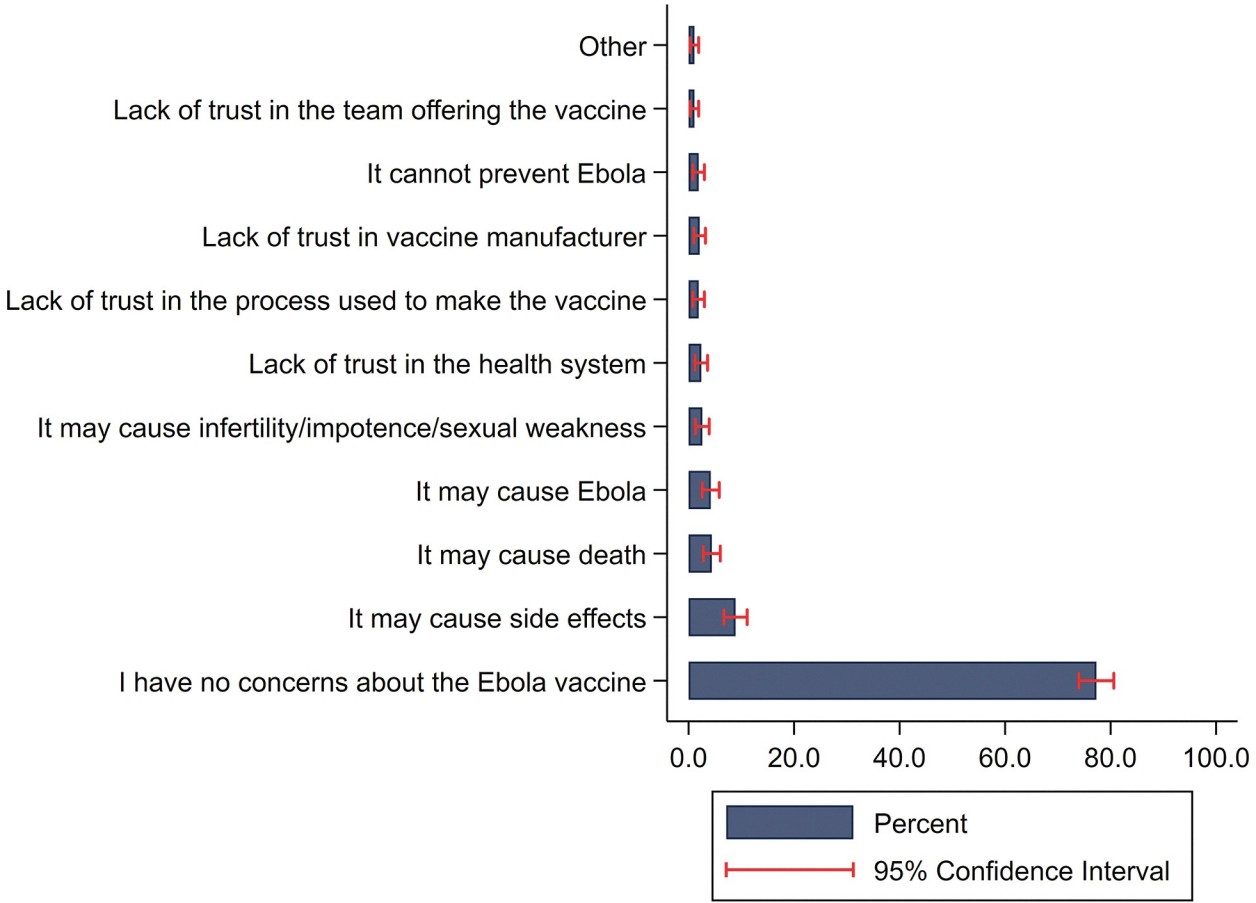

**Fig 1. Percent distribution of reported concerns regarding Ebola vaccines among individuals aware of the vaccine at the time of the interview (N = 620).**

at the time of the survey. Greater confidence in health services to treat EVD in the high-risk districts may have been influenced by the increased investments in the overall health systems in these high-risk districts as part of outbreak preparedness and response. For example, during the 2014 EVD outbreak in Sierra Leone, districts with active EVD outbreaks had intensified social mobilization, greater infrastructure and response capacity (e.g. additional ambulances), and district response centers [20]. Our finding of higher confidence in health services to treat Ebola in high-risk districts is consistent with results from Sierra Leone where respondents who lived in districts with active Ebola outbreaks in 2015 expressed more confidence in the health care system to treat Ebola and provide other health services compared to nonactive districts [20].

Lower confidence in health services to treat EVD compared to malaria and tuberculosis may be reflective of the higher current and historical prevalence of these diseases in Uganda and participants' more recent experiences with health systems for these conditions. Confidence in health services to treat Ebola in Sierra Leone in 2015 was substantially higher compared to our study population (93% vs 55%) [20], but our ability to directly compare our results with settings that had active Ebola transmission at the time of data collection is limited. This suggests that, if an Ebola outbreak originates or spills into low-risk districts, RCCE to strengthen public confidence in Ebola treatment services needs to be rapidly scaled up given the current lower baseline confidence compared to the high-risk districts. RCCE messages

**Table 4. Confidence in health services to treat Ebola, TB and malaria, by sociodemographic characteristics, Uganda, March 2020.**

| Characteristic | Confident in health services to treat Ebola[†] | | | Confident in health services to treat TB[††] | | | Confident in health services to treat Malaria[†††] | | |
|---|---|---|---|---|---|---|---|---|---|
| | N = 3,302 | | | N = 3,195 | | | N = 3,460 | | |
| | N | % | aPR (95% CI) | N | % | aPR (95% CI) | N | % | aPR (95% CI) |
| **Sex** | | | | | | | | | |
| Female | 1,997 | 54.5 | Reference | 2,017 | 75.7 | 0.98 (0.94, 1.02) | 2,090 | 93.0 | Reference |
| Male | 1,305 | 55.1 | 0.96 (0.89, 1.04) | 1,313 | 79.9 | | | 1,370 | 93.1 | 0.99 (0.97, 1.01) |
| **Head of household** | | | | | | | | | |
| No | 1,615 | 52.6 | Reference | 1,636 | 74.4 | Reference | 1,702 | 93.2 | Reference |
| Yes | 1,687 | 56.8 | 1.04 (0.96, 1.14) | 1,694 | 79.7 | 1.00 (0.96, 1.05) | 1,758 | 92.7 | 0.99 (0.98, 1.01) |
| **Age (years)** | | | | | | | | | |
| 15–24 | 710 | 56.0 | 1.05 (0.92, 1.20) | 719 | 76.2 | 0.97 (0.90, 1.04) | 753 | 92.3 | 0.99 (0.97, 1.03) |
| 25–34 | 844 | 50.6 | 1.03 (0.91, 1.16) | 839 | 71.1 | 0.97 (0.92, 1.02) | 870 | 94.2 | 1.00 (0.97, 1.03) |
| 35–44 | 680 | 53.9 | 1.01 (0.90, 1.14) | 686 | 73.8 | 0.96 (0.91, 1.01) | 713 | 92.7 | 0.99 (0.96, 1.03) |
| 45–59 | 661 | 58.2 | 1.03 (0.93, 1.14) | 652 | 84.4 | 1.00 (0.95, 1.06) | 682 | 91.2 | 0.98 (0.95, 1.02) |
| 60 or older | 407 | 58.8 | Reference | 434 | 87.9 | Reference | 442 | 94.9 | Reference |
| **Religion** | | | | | | | | | |
| Christian [A] | 2,972 | 55.7 | Reference | 2,989 | 79.0 | Reference | 3,108 | 93.5 | Reference |
| Muslim | 328 | 49.4 | 0.94 (0.84, 1.06) | 339 | 66.3 | 0.97 (0.91, 1.03) | 350 | 90.2 | 0.96 (0.93, 1.00) |
| **Education** | | | | | | | | | |
| No formal education | 612 | 61.9 | Reference | 617 | 84.5 | Reference | 637 | 92.1 | Reference |
| Some primary | 1,642 | 54.1 | 0.99 (0.91, 1.08) | 1,668 | 74.2 | 0.98 (0.94, 1.03) | 1,732 | 92.6 | 0.99 (0.97, 1.02) |
| Some secondary or higher | 1,043 | 53.2 | 0.99 (0.89, 1.09) | 1,040 | 78.1 | 1.02 (0.97, 1.07) | 1,086 | 93.6 | 1.01 (0.98, 1.04) |
| **Residential setting** | | | | | | | | | |
| Urban | 1,043 | 52.2 | Reference | 1,048 | 72.9 | Reference | 1,080 | 92.5 | Reference |
| Rural | 2,259 | 58.6 | 1.09 (0.98, 1.22) | 2,282 | 83.5 | 1.06 (0.99, 1.14) | 2,380 | 93.8 | 1.02 (0.99, 1.05) |
| **District** | | | | | | | | | |
| Low Ebola risk | 1,109 | 48.2 | Reference | 1,183 | 83.8 | Reference | 1,216 | 93.6 | Reference |
| High Ebola risk | 2,193 | 56.0 | 1.22** (1.08, 1.38) | 2,147 | 75.7 | 0.99 (0.93, 1.05) | 2,244 | 92.9 | 0.99 (0.97, 1.03) |

N denotes number of respondents (un-weighted), % percentage estimates and CI confidence interval both weighted for survey sampling

* p< 0.05

**p<0.01; Design-based F-statistic p values comparing prevalence across categories of a specific characteristic.

† Of the total respondents in the survey (N = 3,485), this analysis excluded: n = 133 responded never heard of Ebola, n = 50 declined to respond.

†† Analysis excluded: those responded never heard of TB (n = 116), declined to respond (n = 39).

††† Analysis excluded: those responded never heard of malaria (n = 12), declined to respond (n = 13).

‖ Missing values: religion (n = 2), education (n = 5)

¶ Age categorized and residential setting classified as per the 2016 Uganda Demographic and Health Survey provided by the Uganda Bureau of Statistics

should describe ETUs and the specific treatment services they provide, emphasizing how available services can significantly improve chances of survival if given early.

Vaccine confidence and perceptions are shaped by many factors, including confidence in the health systems that deliver them. While half of the respondents did not have confidence in health services to treat Ebola in their district, it is possible that confidence in other health services bolstered trust in vaccines and vaccinators. In the event of an outbreak, intensified efforts may be needed to strengthen confidence in health services to treat Ebola in order to effectively detect, treat, and isolate EVD cases. While consistent community engagement activities before, during, and after outbreaks are needed to build long-term trust in health systems, our findings suggest vaccination efforts can be a successful pillar of EVD prevention and control even in

settings with low trust in health services to treat health outcomes. Nearly 5000 healthcare workers were vaccinated in 165 facilities in Uganda [5]. During the recent 2022 outbreak of Sudan ebolavirus [21] in the previously low-risk district of Mubende, Ugandan health officials rapidly disseminated RCCE messages to explain the purpose of the ETUs and to build community trust. A ring vaccination protocol (Solidarity-Tokomeza Ebola) was developed and locally approved to test three candidate vaccines. However, because Sudan ebolavirus vaccines were only available through a clinical trial, in contrast to the availability of licensed vaccines for Zaire ebolavirus, RCCE for Sudan ebolavirus vaccines was very limited.

## Limitations

Our findings are susceptible to social desirability bias, and individuals in high-risk districts where RCCE messages about Ebola were more widely disseminated may have been more likely to provide positive views about the vaccine and health services [17]. Nevertheless, our findings of high confidence for Ebola vaccines in high-risk areas for EVD outbreaks are consistent with population-based studies conducted in South Sudan, Guinea, and Sierra Leone [9, 12]. We were unable to examine predictors of vaccine confidence in our study population given the low proportion of respondents that were aware of vaccines and low heterogeneity in perceived need, perceived acceptability, and vaccination intentions. KAP studies of Ebola vaccine confidence should be a standard part of EVD preparedness activities to inform vaccine-specific communications and to identify groups with low confidence that require additional focus. We did not comprehensively measure vaccine confidence and may not have captured important variability in other domains of confidence, such as the belief that vaccines work, trust in vaccine safety, and perceived benefits of vaccination. Future qualitative studies can help better understand vaccine confidence as well as perceptions about Ebola services. Early and ongoing information sharing about the importance, effectiveness and safety of Ebola vaccines will not only build vaccine confidence but will help establish early support for the vaccination teams and vaccination activities should an outbreak occur.

## Conclusion

Our findings suggest that public confidence in Ebola vaccines was strong once people had received information about them, which may yield high vaccination uptake in the event of an EVD outbreak. Ongoing risk information, education, and communication about Ebola and community engagement (RCCE) that highlight the critical roles of communities in supporting care and immunization efforts during periods without outbreaks may be a critical strategy for strengthening Ebola emergency preparedness and response in the event of an outbreak.

## Supporting information

**S1 Checklist. Inclusivity in global research checklist.**
(DOCX)

**S1 Table. Awareness of Ebola virus disease, tuberculosis, and malaria by sociodemographic characteristics, Uganda, March 2020.**
(DOCX)

**S2 Table. Perceptions about Ebola vaccines and confidence in health services by district-level Ebola risk profile, Uganda, March 2020.**
(DOCX)

**S3 Table. Sociodemographic characteristics by people who were and were not aware of Ebola vaccines, Uganda, March 2020.**
(DOCX)

## Acknowledgments

We thank the study participants, without whom this work would not be possible.

**Disclaimer:** The findings and conclusions in this report are those of the authors and do not necessarily represent the official position of the U.S. Centers for Disease Control and Prevention.

## Author Contributions

**Conceptualization:** Rosalind J. Carter, Apophia Namageyo-Funa, Victoria M. Carter, Mohammed Lamorde, Tabley Bakyaita, Mohamed F. Jalloh.

**Data curation:** Joseph Musaazi.

**Formal analysis:** Aybüke Koyuncu, Joseph Musaazi.

**Investigation:** Apophia Namageyo-Funa, Victoria M. Carter, Mohammed Lamorde, Rose Apondi, Amy L. Boore, Jaco Homsy, Vance R. Brown, Joanita Kigozi, Maria Sarah Nabaggala, Vivian Nakate, Emmanuel Nkurunziza, Daniel F. Stowell, Richard Walwema, Apollo Olowo, Mohamed F. Jalloh.

**Methodology:** Mohammed Lamorde, Mohamed F. Jalloh.

**Project administration:** Rosalind J. Carter.

**Supervision:** Rosalind J. Carter, Mohammed Lamorde, Dimitri Prybylski, Mohamed F. Jalloh.

**Writing – original draft:** Aybüke Koyuncu.

**Writing – review & editing:** Aybüke Koyuncu, Rosalind J. Carter, Joseph Musaazi, Apophia Namageyo-Funa, Victoria M. Carter, Dimitri Prybylski, Rose Apondi, Mohamed F. Jalloh.

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
