## [Decision Letter · Decision Letter 0]

7 Sep 2023

PGPH-D-23-00582

Public perceptions of Ebola vaccines and confidence in health services to treat Ebola, malaria, and tuberculosis: Findings from a cross-sectional household survey in Uganda, 2020

Dear Dr. Koyuncu,

Thank you for submitting your manuscript to PLOS Global Public Health. After careful consideration, we feel that it has merit but does not fully meet PLOS Global Public Health’s publication criteria as it currently stands. Therefore, we invite you to submit a revised version of the manuscript that addresses the points raised during the review process.

We look forward to receiving your revised manuscript.

Kind regards,

Jianhong Zhou

Staff Editor

Journal Requirements:

3. In the online submission form, you indicated that "The underlying data set cannot be made publicly available because it contains human research participant data; however, it can be made available to any interested researchers upon request. The Centers for Disease Control and Prevention is responsible for approving such requests. To request data access, one must write to the Secretary to the Committee/Head of the corresponding author (Aybüke Koyuncu) mentioning the intended use for the data, contact information, a research project title, and a description of the analysis being proposed as well as the format it is expected. The requested data should only be used for the purposes related to the original research or study". All PLOS journals now require all data underlying the findings described in their manuscript to be freely available to other researchers, either 1. In a public repository, 2. Within the manuscript itself, or 3. Uploaded as supplementary information.

Additional Editor Comments (if provided):

Reviewers' comments:

Reviewer's Responses to Questions

**Comments to the Author**

1. Does this manuscript meet PLOS Global Public Health’s publication criteria? Is the manuscript technically sound, and do the data support the conclusions? The manuscript must describe methodologically and ethically rigorous research with conclusions that are appropriately drawn based on the data presented.

Reviewer #1: Yes

Reviewer #2: Yes

Reviewer #3: Partly

Reviewer #4: Yes

2. Has the statistical analysis been performed appropriately and rigorously?

Reviewer #1: No

Reviewer #2: Yes

Reviewer #3: No

Reviewer #4: Yes

3. Have the authors made all data underlying the findings in their manuscript fully available (please refer to the Data Availability Statement at the start of the manuscript PDF file)?

Reviewer #1: No

Reviewer #2: Yes

Reviewer #3: Yes

Reviewer #4: No

4. Is the manuscript presented in an intelligible fashion and written in standard English?

Reviewer #1: Yes

Reviewer #2: Yes

Reviewer #3: Yes

Reviewer #4: Yes

5. Review Comments to the Author

Reviewer #1: line 127,Define each stage of the study. You used multi-stage sampling, what the first stage consisted of, and so on.

line 130, we need to define how we calculated the size of the sample. what is the prevalence of the event under study?

line 200, I think that Poisson regression is not appropriate for this study. Poisson regression is a prediction model that applies when the target variable Y is a count variable (number of occurrences of an event during a period of time). I don't think the dependent variable here is a count variable. I think we need to use binary logistic regression.

This then involves revising the methodology, results and discussion.

Concerning table 2, I think that there are modalities that are missing from the workforce, such as disagree, somewhat agree, etc.

Reviewer #2: Great work. I think this manuscript will add great value to future vaccination strategies implemented in Uganda.

Solid introduction. The second paragraph provides great contextual information about EVD outbreak in Uganda. Please consider making this your first paragraph and moving the current first paragraph after it or later in the introduction section. The authors tend to switch between ‘Ebola vaccines’ and ‘EVD vaccines’. Consider sticking with one terminology. In the third paragraph, the authors touched upon how vaccine hesitancy may be different in Uganda compared to other Western African countries. Perhaps diving more into this would add more depth to the introduction. For example, there are recent Ebola (and COVID) vaccine hesitancy papers for people in the DRC that were published within the past year.

Line 61: WHO acronym is used before providing definition.

Line 65: DRC acronym is used before providing definition.

Results. Number and proportions should be provided whenever stating results. The authors should consider being consistent in the way they report their values as sometimes only percentages are stated (lines 220-221) and other times both the number and proportion are stated (lines 231-233).

Discussion. Great work on the discussion. The manuscript did a great job focusing on Ebola KAP. The discussion could be developed more by providing more historical (in terms of success of previous ring strategies), political, and socioeconomic context and its influence on confidence in health system. Also, adding data on actual vaccination rates in the reported districts would add more substance to the discussion as well.

References. Unable to find reference 5 and reference 20 link did not work.

Reviewer #3: Only a small proportion of the studied population was aware of the Ebola vaccine. In addition, the authors did not compare the sociodemographic characteristics between people who were aware and not aware of Ebola vaccine. The study conclusion about the vaccine confidence drawn from the population aware of the Ebola vaccine can not stand for the whole population. Its significance is limited.

The analyses on the confidence in health services provided limited exciting results either.

In measures part, the authors decleared "Our main outcomes were perceptions of Ebola vaccines and confidence in health services to treat 139 Ebola, malaria and tuberculosis among respondents who reported being aware of Ebola vaccines." which is inconsistent with Table 4, where all respondents were included.

The authors interviewed the participants using three-way classification variables and generated binary variables in the statistical analyses, which may reduce statistical performances.

Reviewer #4: Koyuncu et al. conducted a cross-sectional household survey in six Ugandan districts to obtain perceptions of Ebola vaccines and confidence in health services to treat Ebola. This study holds importance due to the observed insufficient public knowledge about the existence of Ebola vaccines, indicating a necessity for heightened public awareness. Additionally, the limited trust in healthcare services to effectively manage Ebola cases warrants a more in-depth examination.

Overall, the manuscript is well written. The methodology is transparent, and the statistics appear sound. Ethics approval was obtained.

I’ve only minor comments:

Fig. 1: The figure resolution needs improvement.

Line 65: Should be 2013-2016 epidemic.

Line 66: Should be over 28,000 cases.

Lines 67-72: Missing reference.

Lines 94-95: Should be 2013-2016 epidemic.

Line 148-149: Should be 2013-2016 epidemic.

Table 2: Similar to Figure 1, it would have been interesting to further investigate why confidence was diminished in health services for Ebola versus malaria and TB (e.g., lack of ETUs to support health services, low confidence in proficient staff to support health services, low confidence in the current treatments available, etc.)

6. PLOS authors have the option to publish the peer review history of their article (what does this mean?). If published, this will include your full peer review and any attached files.

**Do you want your identity to be public for this peer review?** For information about this choice, including consent withdrawal, please see our Privacy Policy.

Reviewer #1: No

Reviewer #2: No

Reviewer #3: No

Reviewer #4: No

---

## [Editor Report · Decision Letter 1]

1 Nov 2023

Public perceptions of Ebola vaccines and confidence in health services to treat Ebola, malaria, and tuberculosis: Findings from a cross-sectional household survey in Uganda, 2020

PGPH-D-23-00582R1

Dear Miss. Koyuncu,

We are pleased to inform you that your manuscript 'Public perceptions of Ebola vaccines and confidence in health services to treat Ebola, malaria, and tuberculosis: Findings from a cross-sectional household survey in Uganda, 2020' has been provisionally accepted for publication in PLOS Global Public Health.

Please note that I was assigned the Guest Editor role after initially serving as a Reviewer for this manuscript. To preserve transparency and uphold the integrity of the scientific review process, I am required to disclose my original role as Reviewer (see comments below).

Best regards,

Courtney Woolsey

Guest Editor

Reviewer Comments (for reference):

Koyuncu et al. conducted a cross-sectional household survey in six Ugandan districts to obtain perceptions of Ebola vaccines and confidence in health services to treat Ebola. This study holds importance due to the observed insufficient public knowledge about the existence of Ebola vaccines, indicating a necessity for heightened public awareness. Additionally, the limited trust in healthcare services to effectively manage Ebola cases warrants a more in-depth examination.

Overall, the manuscript is well written. The methodology is transparent, and the statistics appear sound. Ethics approval was obtained.

I’ve only minor comments:

Fig. 1: The figure resolution needs improvement.

Line 65: Should be 2013-2016 epidemic.

Line 66: Should be over 28,000 cases.

Lines 67-72: Missing reference.

Lines 94-95: Should be 2013-2016 epidemic.

Line 148-149: Should be 2013-2016 epidemic.

Table 2: Similar to Figure 1, it would have been interesting to further investigate why confidence was diminished in health services for Ebola versus malaria and TB (e.g., lack of ETUs to support health services, low confidence in proficient staff to support health services, low confidence in the current treatments available, etc.)